biomaterials/nanotechnology

two-step anodic oxidation, TiO₂ nanotubes, surface properties, corrosion behaviour, bioactivity

**Author for correspondence:**
Jiahua Ni
e-mail: nijiahua2012@gmail.com

This article has been edited by the Royal Society of Chemistry, including the commissioning, peer review process and editorial aspects up to the point of acceptance.

# Surface properties and bioactivity of TiO₂ nanotube array prepared by two-step anodic oxidation for biomedical applications

## Zhaoxiang Peng[1] and Jiahua Ni[2]

[1]Department of Orthopaedic Surgery, Ningbo Medical Treatment Center Lihuili Hospital, Ningbo 315040, People's Republic of China
[2]State Key Lab of Metal Matrix Composites, School of Materials Science and Engineering, Shanghai Jiao Tong University, Shanghai 200240, People's Republic of China

ZP, 0000-0001-6508-4100

A highly ordered TiO₂ nanotube array has been prepared on a commercial pure titanium substrate in a hydrofluoric (HF) electrolyte using a DC power source through two-step anodic oxidation. The morphology, composition, wettability and surface energy of the nanotube array have been characterized by using a field-emission scanning electron microscope (FE-SEM), a transmission electron microscope (JEM-2010) with energy-dispersive X-ray spectrometer EDX (INCA OXFORD), X-ray diffraction method, an atomic force microscope (AFM), an optical contact angle measuring device and the Owens method with two liquids. The electrochemical behaviours of anodic oxidation films with different structures have been investigated in Sodium Lactate Ringer's Injection at $37 \pm 1°C$ by potentiodynamic polarization curve and electrochemical impedance spectroscopy. The formation mechanism of the nanotube array and the advantages of two-step oxidation have been discussed according to the experimental observation and the characterized results. Meanwhile, the structural changes of nanotubes are analysed according to the results of impedance spectroscopy. Cytotoxicity testing and cell adhesion and proliferation have been studied in order to evaluate the bioactivity of the nanotube array film. The diameters of nanotubes are in the range of 120–140 nm. The nanotube surface shows better wettability and higher surface energy compared to the bare substrate. The nanotube surface exhibits a wide passivation range and good corrosion resistance. The growth of the nanotube array is the result of the combined action of the anodization and field-assisted dissolution. The nanotube array by two-step oxidation becomes more regular and

orderly. Moreover, the nanotube array surface is non-toxic and favourable to cell adhesion and proliferation. Such nanotube array films are expected to have significant biomedical applications.

## 1. Introduction

It is often reported that titanium and its alloys are extensively used as implants in orthopaedics and dentistry for their superior mechanical properties, low modulus, excellent corrosion resistance and good biocompatibility. These excellent properties have already brought widespread clinical success. But titanium cannot directly bond to the bone because it is bioinert. The native oxide layer spontaneously formed on titanium is thin (about 3–8 nm), uneven, amorphous and stoichiometrically defective [1–4].

Materials implanted into the human body are hostile and extremely sensitive [5,6]. It is necessary to improve the biological properties of the titanium surface in order to promote the bond between materials and bone. Therefore, a variety of methods have been attempted over the past decades to further improve the coating quality and optimize the biocompatibility of titanium surfaces, such as ion implantation, plasma spray, alkali treatment and pulsed laser deposition [6–11].

Among the methods which improve the interfacial properties and clinical lifetime of Ti-based implants, anodization has attracted great attention because of its controllable, reproducible results and its being a simple process [12–15]. The fabrication of $TiO_2$ nanotube array by anodic oxidation on titanium foils in fluoride-based electrolyte solution was first reported in 2001 by Grimes *et al.* [16]. The anodic oxide layer is porous and highly ordered. Subsequently, most studies focused on the surface morphology and preparation techniques of nanotubes [17–19]. However, the accurate growth mechanism is still in the speculative stage and needs more in-depth studies [20–22]. It has been found recently that nanoscale porous and tubular oxide layers on titanium implants can promote the bioactivity of substrates [23–27]. Interactions between implants and cells mainly depend on surface properties such as topography, composition, surface roughness, wettability and surface energy [28,29]. The physical and chemical properties of these nano-dimensional structures are strongly dependent on their geometrical features such as tube diameter, tube length and wall thickness, etc., which also determine the function of these nanostructured functional materials [30–32]. But, the surface physicochemical properties of implants are very important for the biomedical applications.

In the present work, on the basis of drawing on porous alumina membrane preparation, highly ordered nanotube oxide films have been prepared on commercial pure titanium (CP-Ti) in fluoride-based electrolyte by the two-step oxidation method. The ordering of nanotubes has been improved effectively. The surface properties of the nanotube array, including morphology, composition, roughness, wettability, surface energy and electrochemical behaviours, are investigated. The possible formation mechanism and the structure of nanotube are analysed based on the previous studies and this experiment. And the initial interaction between nanotubes and cells is also observed and discussed.

## 2. Experimental set-up

### 2.1. Material preparation

Commercially pure titanium sheets (99.5% purity), whose element composition is shown in table 1, have been used as anodic oxidation substrates. The samples are ultrasonically cleaned in acetone in order to remove oil pollution and are ground using #200, #600, #800, #1000 and #1500 abrasive papers. Then they are chemically polished for 60 s in the mixture of $HNO_3$ and HF (V($HNO_3$):V(HF) = 1 : 1) and rinsed by deionized water. Electrochemical experiments are carried out using a DC power source. The electrolyte solution consists of 0.6vol% hydrofluoric acid in water, and a stainless steel electrode serves as a cathode. Anodization is performed by increasing the applied voltage from 0 V to the desired potential, followed by holding the sample in the fixed potential for a desired time under stirring conditions. In this paper, the samples are prepared using two-step oxidation. The general process is as follows: titanium is anodized in a short time (about 10–15 min), and then the sample is immersed into the mixture of $HNO_3$ and HF (V($HNO_3$):V(HF) = 1 : 1) again for dissolving the formed film, and then washed by deionized water. Subsequently, the sample is anodized again immediately in the same condition until the desired time.

**Table 1.** The chemical composition of the commercial pure titanium.

| type | contents of impurities (<wt%) | | | | | substrate |
| | Fe | O | C | N | H | Ti |
| --- | --- | --- | --- | --- | --- | --- |
| TA$_1$ | 0.20 | 0.18 | 0.08 | 0.03 | 0.015 | Remains |

The whole process of anodization is completed at 25°C. The anodized samples are washed with distilled water repeatedly and dried with a blower after anodization.

## 2.2. Characterization

The surface, lateral and bottom morphologies as well as the element composition of the oxide films anodized under different oxidation parameters were observed by a field-emission scanning electron microscope (FE-SEM) (FEI SIRION 200) and a transmission electron microscope (TEM) (JEM-2010) with an energy-dispersive X-ray spectrometer EDX (INCA OXFORD). The heating rate was 5°C min$^{-1}$. The phase components of the as-formed and the heat-treated nanotubules were analysed using X-ray diffraction (C/max-2550/PC) with Cu Kα radiation at 35 kV and 200 mA in the range of 20°–80°, a scanning speed of 5° min$^{-1}$ and a step size of 0.02°.

The surface roughness of the before and after anodized samples was measured by a steep instrument (AS500). The scan was used four times with a scan distance of 2 mm at different regions on each sample for obtaining an average roughness value $R$a. The stylus radius was 12.5 μm. The surface profile of the nanotube structure was examined by an atomic force microscope (AFM, Multimode Nanoscope Scanning, Veeco) to observe the surface relief. The force constant and the resonance vibration frequency of the typical tip (NSC611) were 48 N m$^{-1}$ and 330 KHz, respectively. The measurements were conducted in ambient air under tapping mode. The scan area was $2 \times 2$ μm$^2$. AFM data were analysed using the Nanoscopev7.20 software.

To explore the wettability and surface energy of the nanotube surface, the sessile drop method was used for contact angle measurements with an optical contact angle measuring device (OCA20) with a camera. A 4 μl droplet of distilled water or diiodomethane (CH$_2$I$_2$) was dripped from the tip of microliter syringe on the surface of samples and paused 5 s, then the contact angles between the drop and substrate were measured; meanwhile, the images were collected by the camera. At least five measurements were made on different spots for each specimen. The Owens methods with two liquids [33] were employed for calculating the surface energy. The unanodized substrate was used as a control group.

Electrochemical behaviours, including open circuit potential (OCP), potentiodynamic polarization curves (PPC) and electrochemical impedance spectroscopy (EIS), were performed by the Advanced Electrochemical System (PARSTAT2273, Princeton Applied Research) in Sodium Lactate Ringer's Injection (6 g l$^{-1}$ NaCl, 0.3 g l$^{-1}$ KCl, 0.2 g l$^{-1}$ CaCl$_2$, 3.1 g l$^{-1}$ C$_3$H$_5$NaO$_3$) at 37 $\pm$ 1°C, in which the salt concentration corresponded to that of body fluids. A conventional three-electrode system, with Pt electrode as auxiliary electrode and saturated calomel electrode as reference, was used. The samples for the electrochemical study were prepared as described above. The sample edges and backs were carefully covered with epoxy resin to avoid the possible crevice attack and the area of the sample exposed was 1 cm$^2$. The scan rate of potentiodynamic polarization was 5 mV s$^{-1}$ and the scan range was $-1500$–$3000$ mV. The corrosion parameters were determined by tafel extrapolation. EIS experiments were performed with the frequency domain $10^{-1}$ Hz to $10^4$ Hz when OCP reached steady state. Equivalent circuit models were proposed and the acquired data were fitted to theoretical data using ZSimpWin software.

## 2.3. Biological experiments

### 2.3.1. Cytotoxicity testing

L929 cells (mouse fibroblasts) were provided by the Chinese Academy of Sciences (Shanghai) and grown in α-MEM (GIBCO, Grand Island, NY, USA) supplemented with 10% FBS (Hyclone, Tauranga,

New Zealand) and antibiotics (penicillin 100 U ml$^{-1}$, streptomycin 100 μg ml$^{-1}$; Hyclone, Logan, UT, USA) in a 37°C humidified atmosphere with 5% $CO_2$.

According to the ISO standards [34] of *in vitro* cytotoxicity evaluation, L929 cells (10$^3$) were seeded in a 96-well plate with 150 μl testing solution for 48 h, along with a control with cells grown in the α-MEM medium and five duplicates for each sample. The testing solutions were extracted using 1 g of T or NT in 5 ml crude α-MEM for 72 h at 37°C. Succinate dehydrogenase activity was then determined by adding 20 ul of a 5 mg ml$^{-1}$ 3-(4, 5-dimethylthiazol-2-yl)-2,5-diphenyltetrazolium bromide (MTT, M-2128, Sigma) to each well, followed by 3 h of incubation at 37°C. MTT is a yellow tetrazolium salt that is reduced only in living, metabolically active cell mitochondria [34,35]. After reduction, a violet formazan dye is formed and the amount of living cells can be quantitated spectrophotometrically [36,37].

Medium and MTT were removed. The formazan product was then solubilized in 100 μl 0.04 mol l l$^{-1}$ HCl in isopropanol. The amount of this dye was quantified by measuring the absorption at 570 nm using an automated plate reader (Perkin-Elmer). Extracts were rated as severely (less than 30%), moderately (30–60%), slightly (60–90%) or not cytotoxic (greater than 90%) based on the activity relative to control values. In this work, all $TiO_2$ nanotube samples for biological experiments were ultrasonically cleaned for half an hour in the distilled water and were immersed into the distilled water for 24 h to minimize HF electrolyte residues in the interior of nanotubes and crevices among nanotubes.

### 2.3.2. Cell adhesion and proliferation

The experimental cells were mouse $C_3H_{10}T_{1/2}$ fibroblasts with multi-differentiation potential (provided by Cell Resource Center of Chinese Academy of Sciences Institute of Life Sciences). The $C_3H_{10}T_{1/2}$ fibroblasts were inoculated in the 10 cm Petri-dish placed in an incubator by 10% fetal calf serum DMEM medium with $1 \times 10^4$ cells per ml density of a total of 10 ml, 37°C, 5% $CO_2$. After three days, when the cells at the bottom of the dish converged, these cells were digested by 0.25% trypsin and were discontinued with 10% fetal bovine serum DMEM medium; at the same time, the cell suspension with $1 \times 10^4$ ml$^{-1}$ cell density was prepared for experimental studies. Pure titanium samples and nanotube array samples after disinfection were put into a 12-well plate hole, then 1.5 ml cell suspension prepared as mentioned above was put into each hole, repeated in the 3-hole, 37°C, 5% $CO_2$ incubator. Samples were taken out after 6 h and 2.5% glutaraldehyde was added under a stable temperature (4°C) and then characterized by SEM.

Numerical data were analysed using standard analysis of variance techniques. The results were reported as means ± s.d., and statistical significance was considered at $p < 0.05$.

# 3. Results

## 3.1. Microstructure and composition of nanotube oxide layers

Figure 1 shows the morphologies of the nanotube array by one-step and two-step oxidation. It is obvious that the nanotube structure with two-step oxidation (figure 1*b*) is significantly improved in orderliness and uniformity compared to one-step oxidation (figure 1*a*). Relevant reasons will be discussed below. The surface morphologies obtained at different voltages are shown in figure 2. When the applied voltage is 10 V and the oxide time is 50 min, the ordered nanoporous oxide layer with honeycomb structure appears on the titanium substrate (figure 2*a*). The diameters of the pores are in the range of 60–80 nm and the hole wall is about 10 nm. A large area and highly ordered nanotube array is prepared at 20 V and 50 min as shown in figure 2*b* and *c*. Irregular circular nanotubes are separate and hollow, and the gap between nanotubes is obvious. The nanotubes are 120–140 nm in the outer diameter and 90–110 nm in the inner diameter, with a wall thickness of 10–20 nm. However, when the voltage rises up to 30 V, the tubular structure is completely damaged and the irregular porous structure covers the substrate surface instead. The results show that the tube-like structure will not form if the applied voltage is too low or too high and the applied voltage plays an important role in the process of nanotube formation.

The lateral and bottom view images of the nanotubes are shown in figure 3. Straight nanotubes anodized for 50 min are about 200 nm in length (figure 3*a*), and the length will not increase as the time increases. There are some obvious spaced rings in the nanotube walls and this means that the wall thickness is uneven. The reasons behind this phenomenon may be voltage fluctuations in the oxidation process [38]. An in-depth study of this phenomenon will be conducted in the follow-up

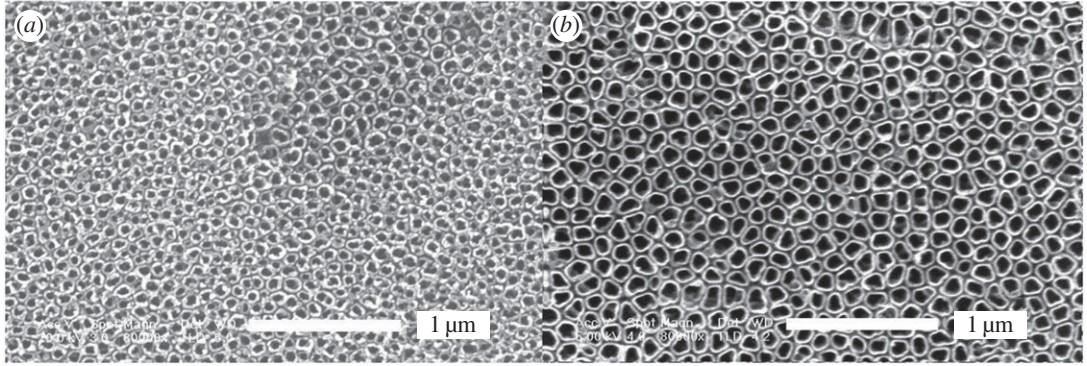

**Figure 1.** SEM images of the nanostructure formed on pure titanium in 0.6vol%HF at 20 V for 50 min. (*a*) one-step oxidation (*b*) two-step oxidation.

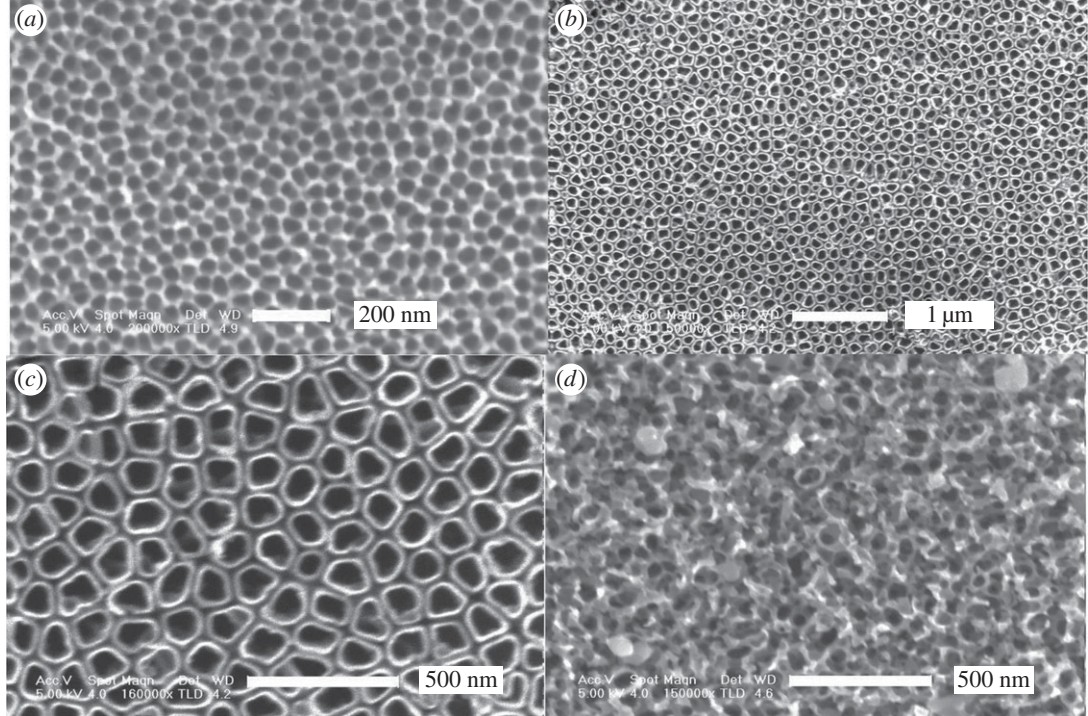

**Figure 2.** SEM images of the nanostructure formed on pure titanium with different applied voltage. (*a*) Honeycomb structure formed in 0.6vol%HF at 10 V for 50 min, (*b*) and (*c*) nanotube arrays formed in 0.6vol%HF at 20 V for 50 min, and (*d*) porous structure formed in 0.6vol%HF at 30 V for 50 min.

experiments. In order to observe the bottom morphology of nanotubes, the film was scratched by a sharp knife resulting in a part of film at the edge of the nick peeling from the substrate. The bottom of the nanotubes and the top of the matrix that removed the nanotubes are visible as shown in figure 3*b* and *c*. The sealed tube bottom indicates that a dense barrier layer formed between nanotubes and the substrate separated the nanotubes from the substrate. On the mating Ti surface (figure 3*c*), there are uniform and evenly distributed concave pits which match the bottom of nanotubes. It is revealed that the bottoms of nanotubes are not flat but spherical. Moreover, there are a few small-sized nanotube bottoms with 20–40 nm in the nanotube array (figure 3*b*), and their presence may be related to the competitive growth mechanism which will be discussed below. From figure 2*b* and *c* and figure 3, it is evident that the nanotubes are open on the top and closed on the bottom, meaning that the nanotube array films could be divided into three layers: nanotubes on the top, dense barrier layer in the middle, titanium substrate at the bottom.

Figure 3*a* shows the TEM images of nanotubes. The gap between the tubes is clearly visible. The electron diffraction pattern of uniform rings but no pots indicates that nanotubes are amorphous (figure 4*b*). Figure 4*d* and table 2 represent the chemical composition of the three selected portions

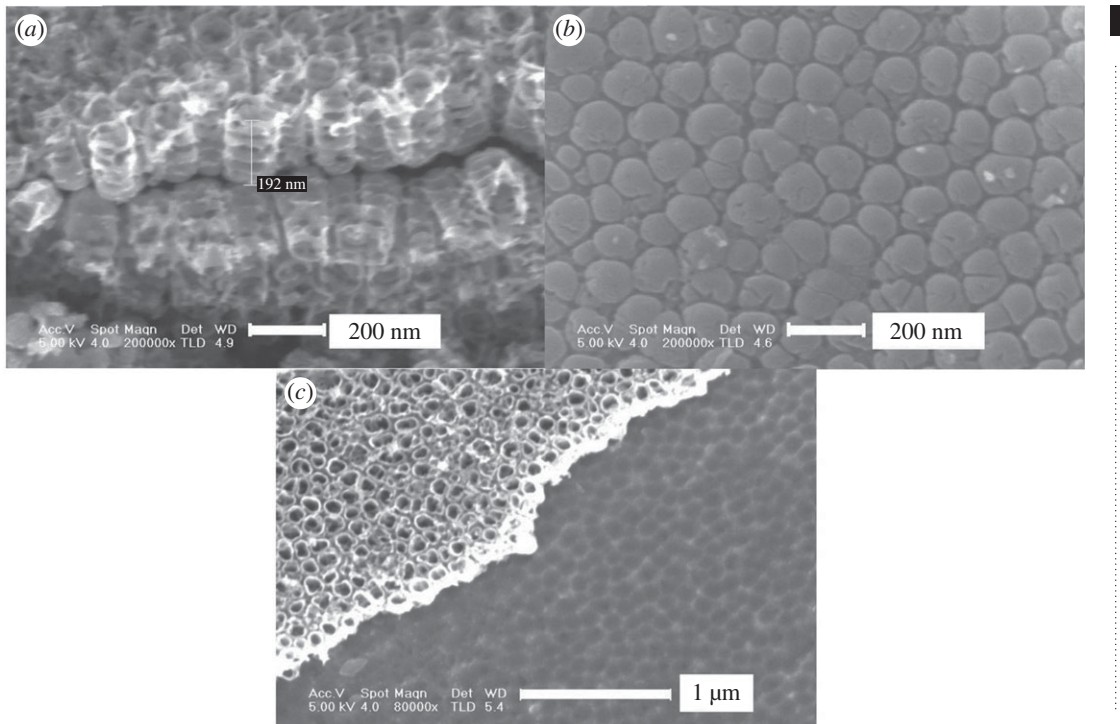

**Figure 3.** SEM images of the nanotube arrays prepared at 20 V in 0.6vol%HF for 50 min. (*a*) lateral view, (*b*) bottom view and (*c*) surface of Ti substrate.

from top to bottom of the nanotubes corresponding to figure 4*c* observed by TEM. The percentage of oxygen atoms (about 60%) is twice that of titanium (about 30%). At the bottom of the nanotubes, the oxygen content decreases and the titanium content increases slightly. Fluorine element from the electrolyte is detected in the nanotubes with an even distribution of about 7–8%. It is revealed that the composition of nanotubes is uniform.

## 3.2. Surface roughness and surface energy of nanotube-like structures

Table 3 shows the average surface roughness of before and after anodized samples. The surface roughness of samples before and after oxidation does not show much difference ($1.02 \pm 0.06$ μm and $0.95 \pm 0.02$ μm, respectively). The nanotube structure does not have much effect on the surface roughness in this experiment.

The representative AFM image of the surface morphology of the nanotube-like structure is shown in figure 5. As shown in figure 5, at the micro-context, TiO$_2$ nanotube array is not in a plane and the surface has ups and downs, which may be related to the surface morphology of Ti-substrate. The rms roughness of the surface was 19.7 nm in the nanoscale.

The sketch maps of contact angle measurement are shown in figure 6. The average contact angle of water on the anodized sample with nanotube arrays is 17.2° and on the Ti-substrate is 86.0°. Such a large difference implies that nanotube arrays significantly improve the wettability of the sample surface. Once the contact angle is over 20° it will be considered to have a surface hydrophobicity. The unanodized Ti-substrate surface shows the hydrophobic nature, and the nanotube array surface displays the hydrophilic nature.

In order to evaluate the surface energy of nanotube arrays, the Owens method was applied [33]. The surface energy can be calculated by the following equation [26]:

$$r_L = r_L^P + r_L^D \, , \tag{3.1}$$

where $r_L$ is the surface energy of liquid and can be decomposed into dispersion force $r_L^D$ and polar force $r_L^P$.

$$r_S = r_S^P + r_S^D \, . \tag{3.2}$$

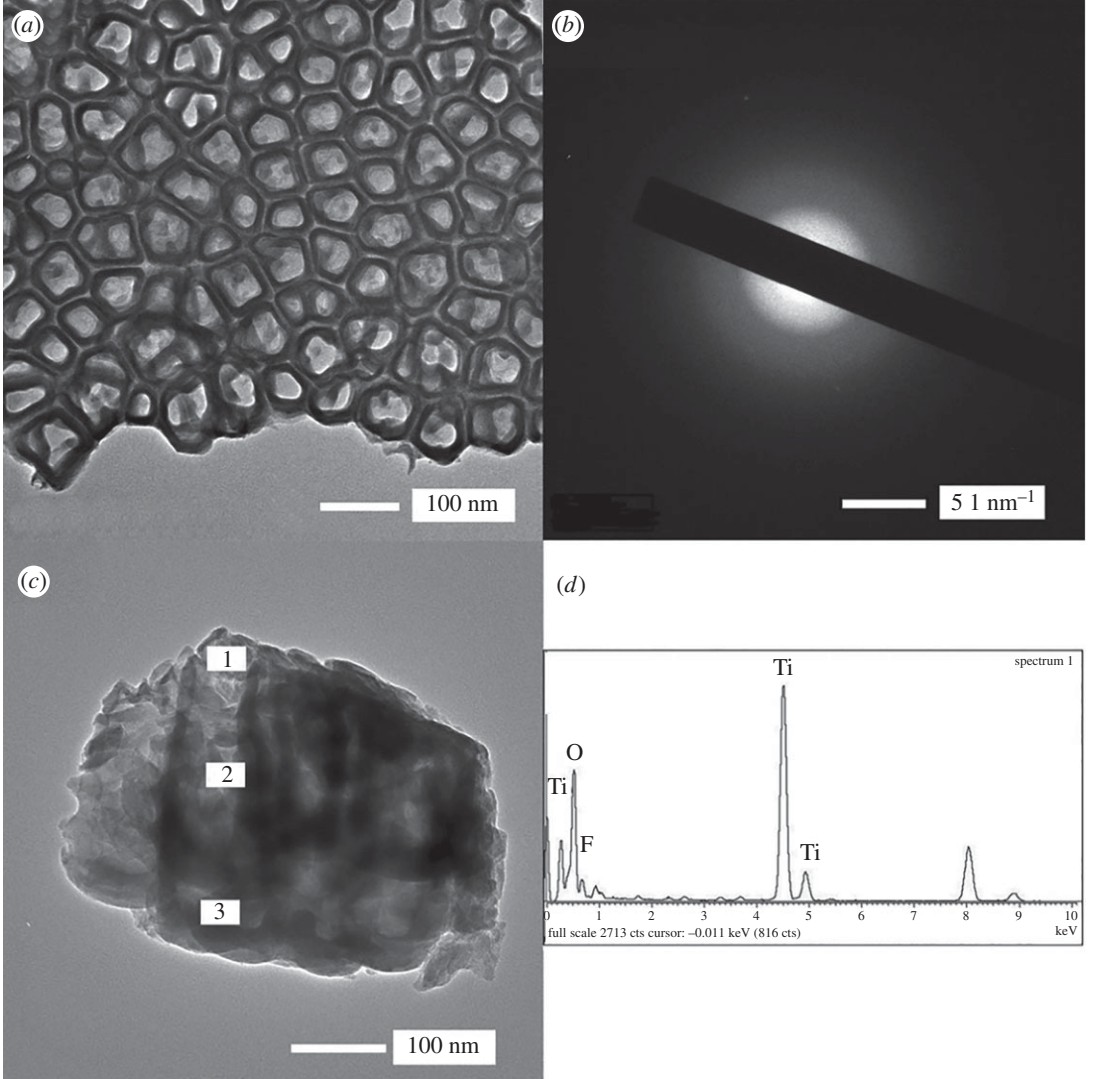

**Figure 4.** (*a*) TEM images of the nanotube arrays prepared at 20 V in 0.6vol%HF for 50 min, (*b*) electron diffraction pattern of nanotubes, (*c*) cross-section nanotubes, selected areas for EDX analysis and (*d*) EDX spectrum of nanotubes.

**Table 2.** TEM/EDX analysis of nanotubes formed on CP-Ti. (All results in at.%).

| spectrum | Ti | O | F |
|---|---|---|---|
| 1 | 31.85 | 61.04 | 7.11 |
| 2 | 30.76 | 61.64 | 7.60 |
| 3 | 38.41 | 53.63 | 7.95 |

**Table 3.** Average surface roughness values for the anodized titanium and non-anodized titanium (μm).

| surface | $R_a$ value |
|---|---|
| anodized Ti | $1.02 \pm 0.06$ |
| non-anodized Ti | $0.95 \pm 0.02$ |

where $r_S$ is the surface energy of solid and also can be decomposed into dispersion force $r_S^D$ and polar force $r_S^D$.

$$r_L(1 + \cos\theta) = 2\sqrt{r_S^D}\sqrt{r_L^D} + 2\sqrt{r_S^P}\sqrt{r_L^P}, \qquad (3.3)$$

where $\theta$ is the contact angle of liquid (*L*) and solid (*S*).

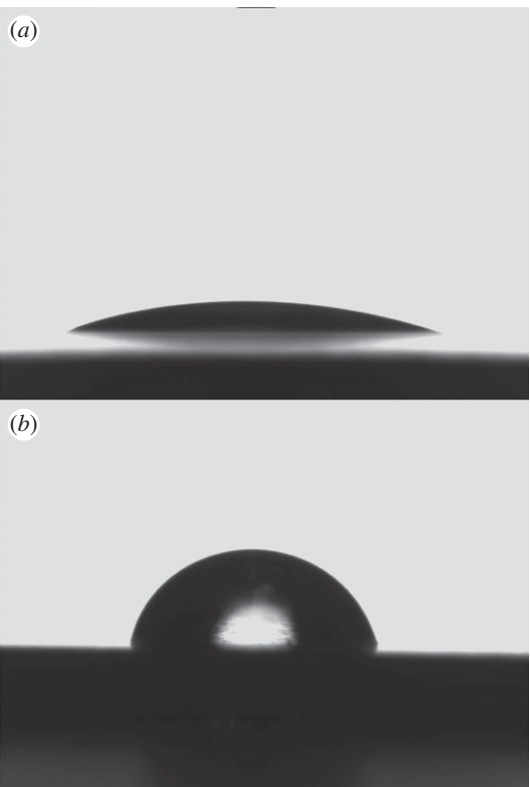

**Figure 5.** Sketch map of measurements of water contact angle (*a*) anodized sample with nanotube structures and (*b*) titanium substrate.

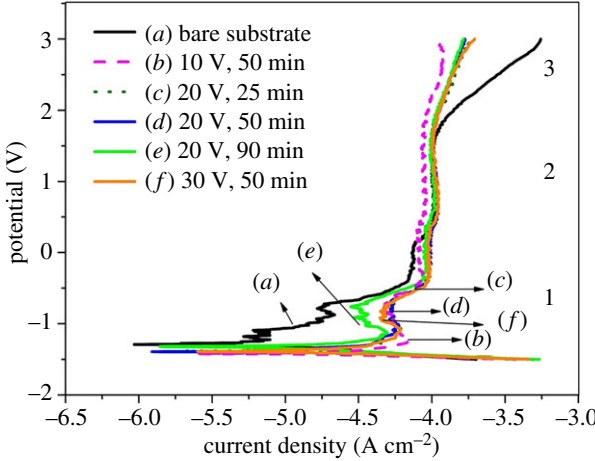

**Figure 6.** PPC of titanium under different oxidation conditions in Sodium Lactate Ringer's Injection at 37 ± 1°C: (*a*) bare substrate; (*b*) applied voltage 10 V, oxidation time 50 min; (*c*) 20 V, 25 min; (*d*) 20 V, 50 min; (*e*) 20 V, 90 min and (*f*) 30 V, 50 min.

According to equations (3.1)–(3.3) and the data in table 3 and table 4 [39], the surface energy of the nanotube arrays and unanodized Ti-substrate can be calculated. The values of the surface energy calculations are summarized in table 5. The surface energy of $TiO_2$ nanotube arrays (about 72.45 mJ m$^{-2}$) is far higher than that of the unanodized surface (about 49.34 mJ m$^{-2}$). A low contact angle led to high surface energy. A good hydrophilicity of nanotube arrays is closely related to their special structure.

## 3.3. Electrochemical investigation of oxide layers

The representative PPC obtained for anodized samples under different oxide conditions are shown in figure 6. The corresponding corrosion parameters are listed in table 6. An immersion period of 2 h has

**Table 4.** Surface energies of the two test liquids (mJ m$^{-2}$) [32].

|  | $r_L^P$ | $r_L^D$ | $r_L$ | $r_L^P/r_L^D$ |
|---|---|---|---|---|
| H$_2$O | 51.0 | 21.8 | 72.8 | 3.26 |
| CH$_2$I$_2$ | 2.3 | 48.5 | 50.8 | 0.05 |

**Table 5.** The surface energy of anodized titanium with TiO$_2$ nanotube arrays and unanodized titanium calculated by the Owens method with two liquids (mJ m$^{-2}$).

|  | $r_S^D \pm \sigma$ | $r_S^P \pm \sigma$ | $r_S \pm \sigma$ |
|---|---|---|---|
| anodized surface | 35.73 $\pm$ 0.3 | 36.72 $\pm$ 0.5 | 72.45 $\pm$ 0.4 |
| unanodized titanium | 49.03 $\pm$ 0.6 | 0.31 $\pm$ 0.2 | 49.34 $\pm$ 0.2 |

**Table 6.** Corrosion parameters from polarization plots.

| samples | $E_{corr}$ (V) | $i_{corr}$ ($\mu$A cm$^{-2}$) | $i_{pass}$ ($\mu$A cm$^{-2}$) |
|---|---|---|---|
| Bare | $-1.274$ | 5.13 | 95.5 |
| 10 V, 50 min | $-1.418$ | 20.4 | 82.2 |
| 20 V, 25 min | $-1.376$ | 6.92 | 93.9 |
| 20 V, 50 min | $-1.394$ | 12.3 | 95.1 |
| 20 V, 90 min | $-1.316$ | 12.5 | 90.3 |
| 30 V, 50 min | $-1.382$ | 14.8 | 95.1 |

been given for stabilization of OCP before starting the experiment. Three different regions are clearly shown, which are marked as 1, 2 and 3 in figure 6. In region 1, the anodized samples (curves (B)–(F)) exhibit active dissolution and active–passive transition behaviour. In the active–passive transition stage, the current fluctuation is relatively large accompanied by significant decreases and increases, which indicates the activation and passivation compete with each other. However, the current density of bare substrate (curve (A)) has no large fluctuation, but increases as the potential increases. In table 6, the bare substrate exhibits significantly higher values of corrosion potential ($E_{corr}$) and lower values of corrosion current density ($i_{corr}$) than anodized films, which suggests that the bare substrate shows better corrosion resistance in the active dissolution region. This may be associated with the thin compact native oxide layer on the bare substrate which spontaneously forms when exposed to air or other oxygen-containing environments [2,3]. In region 2, the current density of each sample shows almost no change with the increase of the potential, indicating the corrosion curves enter into a stable passivation region (region 2) and this region extends over a wide potential. The passive potential is similar (about $-0.39$ V) and only small differences exist in the passive current density ($i_{pass}$) of each sample. It can be inferred that these anodized films with nanoporous, nanotubes and damaged nanotube structures show excellent corrosion resistance because of a wide passive potential range. When the potential reaches 1.62 V (region 3), the current density of bare substrate suddenly increases with the increase of voltage, which may be due to the breakdown of the dense passivation film on bare titanium. Meanwhile, the breakdown does not occur on these anodized films, which suggests that these anodized films have a better corrosion resistance than bare titanium throughout the entire electrochemical process.

To further study the properties of anodized films with nanoporous, nanotube and damaged nanotube structures, electrochemical impedance spectroscopy (EIS) measurements have been carried out. All impedance data are expressed by bode plots which enable equal presentation of all impedance parameters. Figure 9 shows the experimental impedance results of these anodized films under different oxidation conditions in the Sodium Lactate Ringer's Injection, which are measured under open circuit conditions at $37 \pm 1°$C. At low and medium frequencies, the high impedance value of

**Table 7.** Equivalent circuit parameters from figure 5.

| sample | $R_s$ ($\Omega$) | $C_1$ (Fcm$^{-2}$) | $n$ | $R_1$ ($\Omega$cm$^2$) | $C_2$ (Fcm$^{-2}$) | $n$ | $R_2$ ($\Omega$cm$^2$) |
|---|---|---|---|---|---|---|---|
| bare surface | 48.39 | $2.697 \times 10^{-5}$ | 0.943 | $4.563 \times 10^5$ | | | |
| 10 V | 70.44 | $3.421 \times 10^{-5}$ | 0.8694 | $4.047 \times 10^{-6}$ | | | |
| 20 V, 25 min | 55.39 | $1.023 \times 10^{-4}$ | 0.9083 | $3.16 \times 10^5$ | $2.577 \times 10^{-5}$ | 1 | 432.2 |
| 20 V, 50 min | 34.47 | $3.971 \times 10^{-5}$ | 0.8664 | 880.6 | $5.153 \times 10^{-5}$ | 1 | $1.394 \times 10^{-5}$ |
| 20 V, 90 min | 70.11 | $2.263 \times 10^{-5}$ | 1 | 683.6 | $7.309 \times 10^{-5}$ | 0.9057 | $1.728 \times 10^{-5}$ |
| 30 V | 53.74 | $3.1 \times 10^{-5}$ | 0.9666 | 564.2 | $6.097 \times 10^{-5}$ | 0.9767 | $1.125 \times 10^{-5}$ |

anodized films with $10^4$–$10^5$ $\Omega$ cm$^{-2}$ suggests good corrosion resistance. The experimental impedance data are fitted by the theoretical data according to the proposed equivalent circuits (figure 7). The capacitance for curve fitting is evaluated instead by a constant phase element. The simulated values are shown in table 7. Chi-square values of $10^{-1}$–$10^{-2}$ suggest excellent agreement between the experimental and simulated values. In figure 7a, the bare substrate exhibits high capacitive behaviour with one time constant and high $R_1$ value of $4.563 \times 10^5$ (table 7), which indicates a single and compact passive layer formed spontaneously covers the substrate. The phase angle plots of sample with nanoporous surface anodized at 10 V has one extreme value and also shows one time constant (figure 7b). A similar equivalent circuit (figure 11a) is used to fit the experimental data for both the bare and nanoporous surfaces. In this circuit, $C_1$ and $R_1$ represent the capacitance and resistance of film, respectively.

In figure 7c–f, the phase angle plots exhibit two extreme values for samples anodized at 20 V and 30 V, and it is estimated that two time constants control the electrode–electrolyte interface and correspond to two interfaces respectively: the barrier layer with high corrosion resistance and the outer porous layer. The curve of logf–phase angle means that both the inner compact layer and the outer porous layer are contributing to the film growth kinetics [40,41]. According to the information reflected in phase angle plots, another similar equivalent circuit is proposed for surfaces anodized at 20 V and 30 V. In the equivalent circuit (figure 11b), $C_1$ and $R_1$ are assigned for the capacitance and resistance of the outer porous layer, meanwhile $C_2$ and $R_2$ correspond to the capacitance and resistance of the inner barrier layer. $R_s$ represents the solution resistance in two equivalent circuits.

For the sample anodized at 20 V and 25 min, the resistance of outer porous layer ($R_1$) is up to $3.16 \times 10^5$ $\Omega$ and the resistance of barrier layer ($R_2$) is 432.2 $\Omega$. For the sample anodized at 20 V and 50 min, only oxidation time is extended, the resistance of outer and inner layer changed greatly. The resistance of outer porous layer decreased to 880.6 $\Omega$ drastically and the resistance of barrier layer increased to $1.728 \times 10^5$ $\Omega$ sharply. The resistances of the outer porous layer and barrier layer of the sample anodized at 20 V and 90 min are similar to the layer anodized at 20 V and 50 min. From the analysis of EIS, it can be inferred that there are great changes in the structure of the outer and inner layers from 25 min to 50 min and the film structure has not changed much from 50 to 90 min. During the time from 25 min to 50 min, the outer porous layer becomes looser leading to the reduction of the resistance from $3.16 \times 10^5$ $\Omega$ to 880.6 $\Omega$; meanwhile, the growth rate of film is greater than the dissolution rate resulting in a thicker barrier layer from 432.2 $\Omega$ to $1.728 \times 10^5 \Omega$. However, during the oxidation time up to a certain point, the structure of film becomes stable and does not change over time. In this experiment, the film structure is stable from 50 to 90 min. The previous test results, in which the length of TiO$_2$ nanotube arrays anodized in the same way from 50 to 90 min, confirm this inference.

For the sample anodized at 30 V and 50 min, the $R_1$ became lower and the $R_2$ began to reduce, when compared to values for 20 V and 50 min. It can be concluded that the film is looser and the barrier layer thickness declines.

## 3.4. Cytotoxicity

The bare titanium and nanotube arrays are all rated noncytotoxic in the MTT tests in L929 cells. The MTT test for the bare titanium and nanotube arrays in L929 cells reveals that at the saturation of leaching liquor, the cell survival is approximately 100% after 24 h incubation (figure 8).

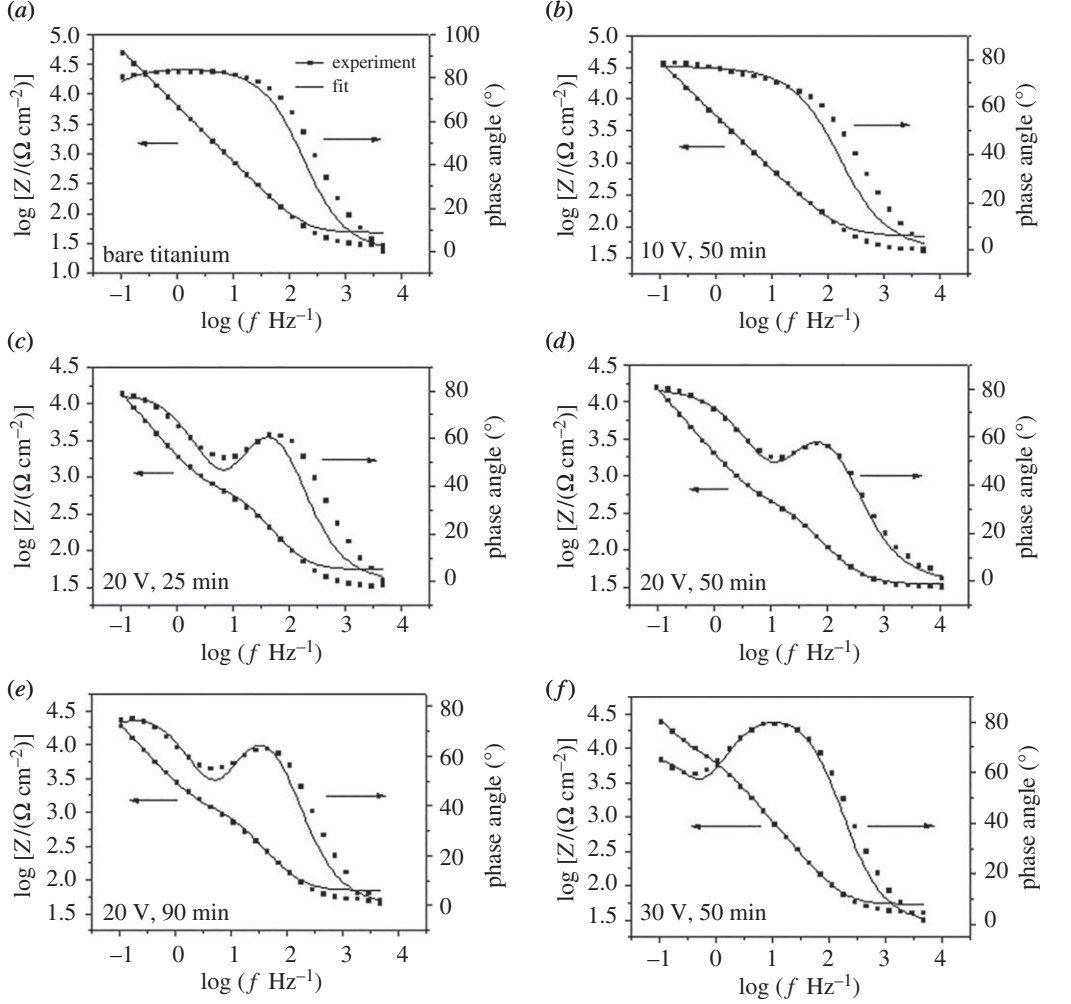

**Figure 7.** Bode plots of titanium under different anodization conditions in Sodium Lactate Ringer's Injection at 37 ± 1°C: (*a*) bare substrate; (*b*) applied voltage 10 V, oxidation time 50 min; (*c*) 20 V, 25 min; (*d*) 20 V, 50 min; (*e*) 20 V, 90 min; (*f*) 30 V, 50 min.

## 3.5. The effect of TiO$_2$ nanotube array on cell adhesion and proliferation

The growth morphology of the C$_3$H$_{10}$T$_{1/2}$ cell cultured for 6 h on Ti and nanotube array surface are shown in figure 9. On the bare titanium (figure 9*a*), the cells are flat-shaped and adhere to the substrate. There is no contact between cells. On the nanotube arrays (figure 9*b*), these cells have already spread out fully and the cell synapse has begun to contact with each other. The cell density per unit area is greater. The nanotube array surface is more favourable for cell growth, which is related to a large extent to the surface properties.

## 4. Discussion

As shown in figure 1*b*, the nanotube structures prepared by two-step anodization are more regular. On the basis of references about the factors affecting the formation of aluminium oxide template [42,43], we consider the possible reasons are as follows: The surface roughness in nanoscale has a great influence on the formation of nanotubes. The larger the roughness, the more uneven the distribution of the electric field, electrolyte and stress on the surface, thus resulting in the initial anodization with the formation of the nanotubes randomly, so the tubes in the face are poorly ordered. The two-step oxidation is based on the result of the first step. If the nanotube array by the one-step oxidation is dissolved, not only can it reduce the surface roughness of the substrate, making the surface more uniform, but also there are many uniform little concave pits after the first-step anodization on the substrate to provide regular template for the second oxidation, hence the nanotube array by the two-step oxidation is more regular.

Based on the observation of surface morphology and the analysis of current fluctuation during oxidation, it is reasonable to assume that the growth process of nanotube arrays could be divided into

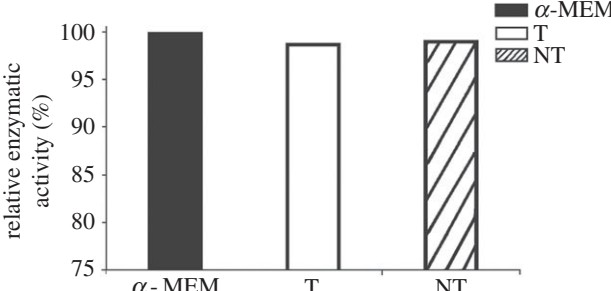

**Figure 8.** MTT-reducing activity relative to control after 48 h exposure of L-929 cells to extracts made from 24 h test samples. Data are means $\pm$ s.d. ($n = 5$).

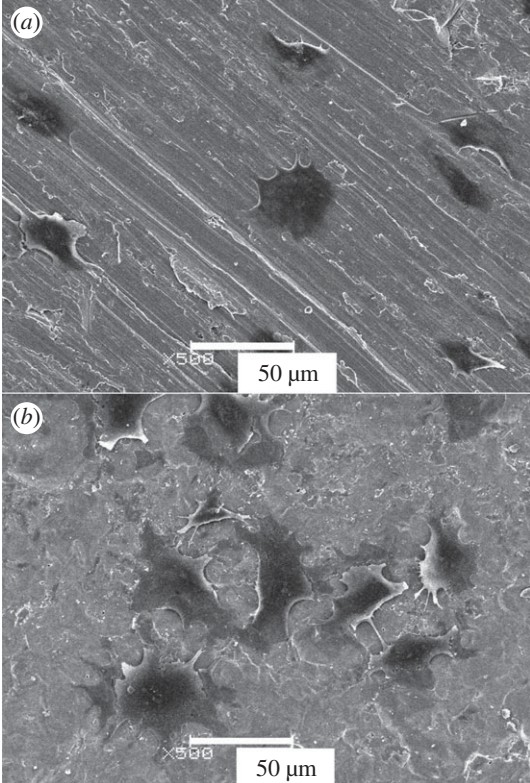

**Figure 9.** C3H10T1/2 cell growth morphology after 6 h culture on Ti (*a*) and nanotube arrays (*b*), respectively.

following three stages: (1) A thin uniform and compact titania film form on bare substrate within a very short period of time after the start of the anodization, in which the sharp decline in current is observed. The formation of the initial oxide film is due to the interaction of $Ti^{4+}$ ions coming from the metal surface with $O^{2-}$ ions from electrolyte. The reaction equation can be described as follows:

$$Ti^{4+} + 2O^{2-} \rightarrow TiO_2. \tag{4.1}$$

After the formation of the initial oxide layer, as a result of electric field, these oxygen anions go through the oxide layer and reach the metal–oxide interface, where they react with the metal; at the same time $Ti^{4+}$ cations move from the metal towards the oxide–electrolyte interface. The compact film gradually thickens. (2) Under the influence of the electric field, the Ti-O bond with high bond energy ($323$ KJ mol$^{-1}$) undergoes polarization and is weakened, thus promoting dissolution of the initial dense layer [44]. Meanwhile, the dissolution of the compact oxide layer appears under the joint action of acidic electrolyte and electric field (also known as field-induced dissolution, rather than a simple chemical dissolution process in the HF electrolyte). The current begins to rise at this stage. The field-induced dissolution of film plays a key role in the formation of nanotube structure [45]. Small

dissolution pits turn up randomly in the relatively weak parts of the film. The localized field-induced dissolution is represented by the following reaction equation:

$$TiO_2 + 6HF \rightarrow [TiF_6]^{2-} + 2H_2O + 2H^+. \tag{4.2}$$

These pits become bigger pores by field-induced dissolution and the pore density increases gradually. Pores are distributed unevenly in the dense film. The structures with external porous and internal compact layer form preliminarily [46].

As the oxygen ions keep on migrating towards metal–oxide interface, the reactions of $Ti^{4+}$ cations and oxygen ions happen and the barrier layer moves towards the substrate continuously, while the pores become deeper and bigger by field-induced dissolution. If the formation rate of the barrier is faster than the dissolution rate, the thickness of the barrier layer increases, on the contrary decreases. As the pores become deeper, the electric field in those protruded regions between pores becomes larger leading to enhanced field-induced dissolution, so the abruption occurs between two pores. The nanotube structure has basically formed. At this point, the rising current begins to stabilize. (3) We believe that the growth mechanism of competition exists in the process of nanotube growth. The growth competition leads to the merger between small neighbouring tubes and big tubes swallowing small tubes, so that it encourages nanotubes to become uniform. This viewpoint is supported by the SEM images of nanotubes (figure 1). There are small tubes and tube wall not completely dissolved in big tubes, which reflects the process of two or several small tubes merging into a big tube and the middle tube wall is not fully dissolved. Finally, when the middle tube wall is fully dissolved, two or more small tubes combine into a big tube. When the growth competition among nanotubes achieves dynamic balance with the ongoing anodization, the structure of individual nanotubes is better ordered by a self-adjustment process.

The field-induced dissolution and generation of the oxide layer in acidic electrolyte take place simultaneously. As the oxidation time extends, when the growth rate of barrier layer at the metal–oxide interface, the field-induced dissolution rate of the oxide layer at the pore bottom–electrolyte interface and the field-induced dissolution rate of the porous layer at the top surface of tubes become equal, the formation and dissolution of the barrier layer and porous layer achieve dynamic balance [47], thus the thickness of the nanotube array and the barrier layer will stop growing and remain within a certain range. After the balance between the formation and dissolution of the layer, the extension of oxide time does not change the structures of the nanotubes and barrier layer, which are supported by the analysis of EIS.

At lower voltage, only nanopores form (figure 2a). The possible reason is that the field-induced dissolution does not take place at these protruded regions among pores, because the electric field strength distributed at these protruded regions is less than the critical value of titania dissolution. In contrast, at higher voltage, the nanotube array is seriously damaged by the high electric field strength and the surface structure become looser and extremely irregular (figure 2d).

It is essential that the optimal surface characteristics should be discovered to potentially mitigate any negative effects of the bulk material on bone formation and function. For example, a sufficient thick oxide layer can restrain corrosion and release harmful compounds. Thompson & Puleo [48] suggested that ions released from implants could impair normal bone formation. Surface roughness and composition are considered to influence the properties of adherent cells [49,50]. Based on these viewpoints proved by these reported studies and combined with the findings of this experiment, the $TiO_2$ film with nanotube array has significant advantages in its structure and composition for biomedical applications. On the one hand, the inner compact barrier layer can effectively prevent titanium ions from releasing into the surrounding body fluids and inhibit the corrosion of body fluids to bare substrate. The outer special structure with regular nanotube arrays improves surface wettability and increases surface energy, which can promote the adhesion and growth of cells. Rough surface is prone to providing anchor points to the cell attachment, thus mechanical interlocking form between the cells and implants resulting in enhancing the cell adhesion. On the other hand, the film contains fluorine through EDX detection. As one of the necessary trace elements in the body for life activity, fluorine has an important influence on bone growth and development of the whole body and the maintenance of bone structure and physiological functioning [51]. Theoretically, fluorine-containing nanotube films will promote the formation of bone on the surface. Additionally, MTT tests confirm that the amorphous $TiO_2$ nanotube film in this paper is noncytotoxic. In the bone implant materials, $TiO_2$ nanotube array on titanium surface has enormous potential for biomedical applications.

# 5. Conclusion

(1) Well-ordered nanotubulars with diameters of 120–140 nm are prepared on commercial pure titanium using DC power source in the HF solution by two-step anodic oxidation. The nanotube arrays by two-step anodic oxidation exhibit a better ordering than the one-step oxidation. The oxide voltage is an important factor for the formation of nanotubes. There are fluorine elements in film except for titanium and oxygen. The composition of amorphous nanotubes is uniform from the top to the bottom. Anodized to a certain time, the structure of nanotubes will become stable.

(2) The wettability and surface energy of nanotube surfaces are significantly higher than those of bare titanium, but the roughness has no much change before and after oxidation. Moreover, the electrochemical studies indicated the anodized surfaces with nanoporous nanotubes and destroyed nanotubes possess better corrosion resistance with a wide passivation range than bare titanium.

(3) The vitro biological tests prove that nanotube array surfaces are non-toxic and favourable for the growth of fibroblasts. The cells on nanotubular array surfaces exhibit greater adhesion, extensibility and proliferation compared to those cultured on bare titanium.

Data accessibility. All the data are available in the manuscript and electronic supplementary files.

Authors' contributions. Z.P. and J.N. designed the study, prepared all samples for analysis, interpreted and wrote the paper. Z.P. collected and analysed the data. Both of the authors gave final approval for publication.

Competing interests. We declare we have no competing interests.

Funding. This research was supported by a grant from the National Natural Science Foundation of China (81401819), Programs Supported by the Ningbo Natural Science Foundation (2018A610203).

Acknowledgements. We thank Prof. H.W. Wang from the Institute of Ecology and Environment Materials of Shanghai Jiao Tong University for providing the electrochemical experimental equipment.

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
