## [Reviewer comments · Royal Society Open Science]

Review History

RSOS-181948.R0 (Original submission)

Review form: Reviewer 1

Is the manuscript scientifically sound in its present form?

Yes

Are the interpretations and conclusions justified by the results?

Yes

Is the language acceptable?

Yes

Is it clear how to access all supporting data?

Not Applicable

Do you have any ethical concerns with this paper?

No

Have you any concerns about statistical analyses in this paper?

No

Recommendation?

Major revision is needed (please make suggestions in comments)

Comments to the Author(s)

As the authors claimed, the Ti-based materials are common materials for the implant of an artificial limb and joint. How to improve the biocompatibility is a critical and practical issue. The authors proposed modified the surface with anode TiO₂ nanotube array to change the surface roughness, wettability and composition to enhance the adhesive of the cell and biocompatibility. It is worth to be investigated.

Please address the following questions

1. pay more attention to the format and the consistency, Some part is Figure some are Fig. In section 3.3 it should be Figure 6, not Figure 8. The table need capitalize the first letter, and so on.
2. The authors need to reorganize the whole paper. What is the critical issue to affect the biocompatibility for the TiO₂ nanotube array modified Ti materials, for example, morphology, roughness, or wettability, and then the paper should focus on these conditions. If the factor is not a critical one, put the corresponding data into supporting information. Otherwise, it will influence the story.
3. Prof. Grimes had been work on the TiO₂ nanotube array a lot. I do not think this is the first time report, (I do not check the literature, I believe it is not)
4. Making the connection between the critical factor and the biocompatibility. Such as the wettability, if the hydrophilic surface is a benefit to the cell growth, show the data, the cell growth depends on the hydrophilic properties.

Review form: Reviewer 2

Is the manuscript scientifically sound in its present form?

No

Are the interpretations and conclusions justified by the results?

No

Is the language acceptable?

Yes

Is it clear how to access all supporting data?

Not Applicable

Do you have any ethical concerns with this paper?

No

Have you any concerns about statistical analyses in this paper?

No

Recommendation?

Reject

Comments to the Author(s)

In this manuscript, the authors claimed that the highly ordered nanotube oxide films have been prepared on commercial pure titanium (CP-Ti) in fluoride-based electrolyte by two-step oxidation method for the first time, and it is non-toxic and favorable for the growth of fibroblasts. However, due to the following problems I think it in the present form is not suitable to be published in the journal.

- 1) It is not true to mention “up to date, there are few reports in the current literatures about the surface roughness, wettability and surface energy of the nanotubular oxide film” or “the exploration of the electrochemical corrosion behavior of the TiO₂ nanotube array film is also limited.” in introduction. Actually, many relevant works have been extensively done in the literature.
- 2) The preparations of the highly ordered nanotube oxide films on titanium in fluoride-based electrolyte by two-step oxidation method have been previously published.
- 3) The experiment seems not sufficient to evaluate the bioproperties of the prepared nanotube oxide films on Ti. More experimental data on biocompatibility are needed.
- 4) Why the highly ordered nanotube oxide films on titanium exhibit excellent adhesion, extensibility, and proliferation compared to those cultured on bare titanium?
- 5) The corrosion current and AC impedance are associated with real surface area, it is hard to compare the electrochemical behavior because the real surface area is different for the different surface treatments.
- 6) The relevant previously published works have not well cited.

Review form: Reviewer 3

Is the manuscript scientifically sound in its present form?

No

Are the interpretations and conclusions justified by the results?

Yes

Is the language acceptable?

Yes

Is it clear how to access all supporting data?

Not Applicable

Do you have any ethical concerns with this paper?

No

Have you any concerns about statistical analyses in this paper?

No

Recommendation?

Accept with minor revision (please list in comments)

Comments to the Author(s)

Pore structure are not uniform in the microscope images. The pores are slightly distorted. More explanation is required.

Fluorine is good for life activity. The source of Fluorine is HF. HF is very toxic. How did authors

make sure of complete removal of HF from TiO₂ nanotubes?

The adhesion property of TiO₂ is not clearly explained in the manuscript.

It would be more interesting if authors compare the wettability and adhesion property of amorphous TiO₂ with crystalline TiO₂ nanotube.

Decision letter (RSOS-181948.R0)

08-Feb-2019

Dear Dr Peng:

Title: Surface properties and bioactivity of TiO₂ nanotube array prepared by two-step anodic oxidation for biomedical applications

Manuscript ID: RSOS-181948

The editor assigned to your manuscript has now received comments from reviewers. We would like you to revise your paper in accordance with the referee and Subject Editor suggestions which can be found below (not including confidential reports to the Editor). Please note this decision does not guarantee eventual acceptance.

Please submit your revised paper before 03-Mar-2019. Please note that the revision deadline will expire at 00.00am on this date. If we do not hear from you within this time then it will be assumed that the paper has been withdrawn. In exceptional circumstances, extensions may be possible if agreed with the Editorial Office in advance. We do not allow multiple rounds of revision so we urge you to make every effort to fully address all of the comments at this stage. If deemed necessary by the Editors, your manuscript will be sent back to one or more of the original reviewers for assessment. If the original reviewers are not available we may invite new reviewers.

On behalf of the Subject Editor Professor Anthony Stace and the Associate Editor Professor Claire Carmalt.

RSC Associate Editor:
Comments to the Author:
(There are no comments.)

RSC Subject Editor:
Comments to the Author:
(There are no comments.)

Reviewers' Comments to Author:
Reviewer: 1

Comments to the Author(s)

As the authors claimed, the Ti-based materials are common materials for the implant of an artificial limb and joint. How to improve the biocompatibility is a critical and practical issue. The authors proposed modified the surface with anode TiO₂ nanotube array to change the surface roughness, wettability and composition to enhance the adhesive of the cell and biocompatibility. It is worth to be investigated.

Please address the following questions

1. pay more attention to the format and the consistency, Some part is Figure some are Fig. In section 3.3 it should be Figure 6, not Figure 8. The table need capitalize the first letter, and so on.
2. The authors need to reorganize the whole paper. What is the critical issue to affect the biocompatibility for the TiO₂ nanotube array modified Ti materials, for example, morphology, roughness, or wettability, and then the paper should focus on these conditions. If the factor is not a critical one, put the corresponding data into supporting information. Otherwise, it will influence the story.
3. Prof. Grimes had been work on the TiO₂ nanotube array a lot. I do not think this is the first time report, (I do not check the literature, I believe it is not)
4. Making the connection between the critical factor and the biocompatibility. Such as the wettability, if the hydrophilic surface is a benefit to the cell growth, show the data, the cell growth depends on the hydrophilic properties.

Reviewer: 2

Comments to the Author(s)

In this manuscript, the authors claimed that the highly ordered nanotube oxide films have been prepared on commercial pure titanium (CP-Ti) in fluoride-based electrolyte by two-step oxidation method for the first time, and it is non-toxic and favorable for the growth of fibroblasts.

However, due to the following problems I think it in the present form is not suitable to be published in the journal.

- 1) It is not true to mention “up to date, there are few reports in the current literatures about the surface roughness, wettability and surface energy of the nanotubular oxide film” or “the exploration of the electrochemical corrosion behavior of the TiO₂ nanotube array film is also limited.” in introduction. Actually, many relevant works have been extensively done in the literature.
- 2) The preparations of the highly ordered nanotube oxide films on titanium in fluoride-based electrolyte by two-step oxidation method have been previously published.
- 3) The experiment seems not sufficient to evaluate the bioproperties of the prepared nanotube oxide films on Ti. More experimental data on biocompatibility are needed.
- 4) Why the highly ordered nanotube oxide films on titanium exhibit excellent adhesion, extensibility, and proliferation compared to those cultured on bare titanium?
- 5) The corrosion current and AC impedance are associated with real surface area, it is hard to compare the electrochemical behavior because the real surface area is different for the different surface treatments.
- 6) The relevant previously published works have not well cited.

Reviewer: 3

Comments to the Author(s)

Pore structure are not uniform in the microscope images. The pores are slightly distorted. More explanation is required.

Fluorine is good for life activity. The source of Fluorine is HF. HF is very toxic. How did authors make sure of complete removal of HF from TiO₂ nanotubes?

The adhesion property of TiO₂ is not clearly explained in the manuscript.

It would be more interesting if authors compare the wettability and adhesion property of amorphous TiO₂ with crystalline TiO₂ nanotube.

Author's Response to Decision Letter for (RSOS-181948.R0)

See Appendix A.

Decision letter (RSOS-181948.R1)

26-Mar-2019

Dear Dr Peng:

Title: Surface properties and bioactivity of TiO₂ nanotube array prepared by two-step anodic oxidation for biomedical applications

Manuscript ID: RSOS-181948.R1

It is a pleasure to accept your manuscript in its current form for publication in Royal Society

Open Science. The chemistry content of Royal Society Open Science is published in collaboration with the Royal Society of Chemistry.

On behalf of the Subject Editor Professor Anthony Stace and the Associate Editor Professor Claire Carmalt.

RSC Associate Editor
Comments to the Author:
The authors have made all the relevant corrections and accept as is recommended.

Reviewer(s)' Comments to Author:

Appendix A

February 20, 2019

Dr. Laura Smith
Publishing Editor
Royal Society Open Science

Dear Dr. Laura Smith:

Responding to your request, we have revised our manuscript entitled “Surface properties and bioactivity of TiO₂ nanotube array prepared by two-step anodic oxidation for biomedical applications” (Manuscript ID: RSOS-181948). The referees’ useful remarks and suggestions are much appreciated. We have responded positively to essentially all of the comments and revised the manuscript accordingly as described below.

All changes made in the manuscript have been marked in red font.

Reviewer: 1

1. “pay more attention to the format and the consistency, Some part is Figure some are Fig. In section 3.3 it should be Figure 6, not Figure 8. The table need capitalize the first letter, and so on.”

--- We thank the Reviewer for the suggestion. We have changed Figure 1 (1st line, 1st paragraph, section 3.1) to Fig. 1; changed Figure 3A (1st line, 3rd paragraph, section 3.1) to Fig. 3A; changed Figure 9 (3rd line, 2nd paragraph, section 3.3) to Fig. 9. We have changed Fig. 8 (4th line, 1st paragraph, section 3.3) to Fig. 6. We have changed “table 3 (1st line, 8th paragraph, section 3.2) to “Table 3”, “table 4 (3rd line, 8th paragraph, section 3.2)” to “Table 4”.

2. “The authors need to reorganize the whole paper. What is the critical issue to affect the biocompatibility for the TiO₂ nanotube array modified Ti materials, for example, morphology, roughness, or wettability, and then the paper should focus on these conditions. If the factor is not a critical one, put the corresponding data into supporting information. Otherwise, it will influence the story.”

--- The reviewer gave us useful suggestions. In this work, we discussed mainly the surface physicochemical properties, including morphology,

composition, surface roughness, wettability and surface electrochemical behaviors, which together determine the effect of TiO₂ nanotube surface on cell behaviors.

3. "Prof. Grimes had been work on the TiO₂ nanotube array a lot. I do not think this is the first time report, (I do not check the literature, I believe it is not)."

--- We thank the Reviewer for the suggestion. We have deleted "for the first time" (3rd line, 4th paragraph, section 1).

4. "Making the connection between the critical factor and the biocompatibility. Such as the wettability, if the hydrophilic surface is a benefit to the cell growth, show the data, the cell growth depends on the hydrophilic properties."

--- We thank the Reviewer for the suggestion. We believe that the effect of biomaterial surface on cell behaviors is acted together by various physical and chemical factors in this work. After implantation, water molecules adsorb on the surface of biomaterials in a very short time, and affect the subsequent protein adsorption process. Some studies have been carried out on the effects of hydrophilicity on cell behaviors, including adhesion, migration and differentiation. Most studies support the view that cell adhesion is guided by ECM (extracellular matrix), and hydrophobic surfaces are more beneficial to protein adsorption. Therefore, the maximum number of cell adhesion will appear at a certain degree of wettability, and the equilibrium value prefers to hydrophilicity. And the main purpose of this work is to introduce the physicochemical properties of the surface of TiO₂ nanotube arrays, and the biological experiments just preliminarily proved the biocompatibility of TiO₂ nanotubes.

Reviewer: 2

1. "It is not true to mention "up to date, there are few reports in the current literatures about the surface roughness, wettability and surface energy of the nanotubular oxide film" or "the exploration of the electrochemical corrosion behavior of the TiO₂ nanotube array film is also limited." in introduction. Actually, many relevant works have been extensively done in the literature."

--- We thank the Reviewer for the suggestion. We have deleted these sentences “Up to date, there are few reports in the current literatures about the surface roughness, wettability and surface energy of the nanotubular oxide film. The exploration of the electrochemical corrosion behavior of the TiO₂ nanotube array film is also limited.” (12-15th line, 3rd paragraph, section 1). And we changed the sentence “But, they are very important for the biomedical applications” (15th line, 3rd paragraph, section 1) to “But, surface physicochemical properties of implants are very important for the biomedical applications.” (13th line, 3rd paragraph, section 1)

2. “The preparations of the highly ordered nanotube oxide films on titanium in fluoride-based electrolyte by two-step oxidation method have been previously published.”

--- We thank the Reviewer for the suggestion. We have deleted “for the first time” (3rd line, 4th paragraph, section 1).

3. “The experiment seems not sufficient to evaluate the bioproperties of the prepared nanotube oxide films on Ti. More experimental data on biocompatibility are needed.”

--- We thank the Reviewer for the suggestion. In this manuscript, we mainly discussed the preparation and surface physicochemical properties of anodized TiO₂ nanotubes, and the biological experiments were adopted to preliminarily evaluate the biocompatibility of anodized TiO₂ nanotube arrays. We will discuss the biological properties of TiO₂ nanotube arrays in detail in our subsequent articles.

4. “Why the highly ordered nanotube oxide films on titanium exhibit excellent adhesion, extensibility, and proliferation compared to those cultured on bare titanium?”

--- We thank the Reviewer for the suggestion. The surface roughness of TiO₂ nanotubes is higher than that of pure titanium due to the special tube structure. When the cells adhere to the surface of nanotubes, their filopodia can anchor in the wall of nanotubes and even reach into the tubes, which is more beneficial to early adhesion of cells. Meanwhile, the surface of nanotube exhibits excellent hydrophilicity, and the media is easy to penetrate into the interior of

tubes and the interval between tubes, which provides adequate nutrition for cell growth.

5. "The corrosion current and AC impedance are associated with real surface area, it is hard to compare the electrochemical behavior because the real surface area is different for the different surface treatments."

--- We thank the Reviewer for the suggestion. In my opinion, AC impedance is not related to the real surface area, but is related to the interface structure and the reaction happened at the interface. Because the principle of AC impedance is to apply a sinusoidal signal with small amplitude potential but different frequency near the open circuit potential, and then record the feedback signal. Thus, the difference between the input and output signals is obtained. This is not current information, so it is independent of the real surface area. Different nanotube thicknesses and different tube diameters have effects on the values of different devices of the AC impedance fitting circuit. The pure titanium surface has only a thin and dense oxide film. The current density is related to the true area of the sample. The corrosion current is the total current on the working electrode (sample) measured by electrochemical workstation. To compare the current density of different samples, the real surface area of the working electrode (sample) should be measured. Therefore, the nanotube arrays may have an effect on the current density. The current characteristics of nanotubes may vary under different applied voltages. We define the current density obtained by macro area (the same area of all samples) as the nominal current density. It can be seen that if tube diameter is large, the real area is close to the macro area, the current density is close to the current density of the pure titanium sample. The smaller the diameter of the tube, the larger the real area and the smaller the current density. Therefore, we can judge the structure of the films on pure titanium surface by polarization curves.

6. "The relevant previously published works have not well cited."

--- We thank the Reviewer for the suggestion. In section "Introduction", we have cited 6 relevant previously published works (Ref.[17]-Ref.[22]), and the subsequent reference number has also been adjusted.

Reviewer: 3

1. "Pore structure are not uniform in the microscope images. The pores are slightly distorted. More explanation is required."

--- We thank the Reviewer for the suggestion. In this manuscript, we prepared TiO₂ nanotube arrays on pure titanium surface in HF electrolyte by anodization. The growth of TiO₂ nanotube array is a self-assembly process, and is a dynamic process of continuous growth at the bottom of nanotube array and continuous dissolution of the tube array surface and tube walls. The ordered nanotube arrays can be obtained at a certain voltage (20V in this manuscript), but at a higher or lower voltage, no nanotube arrays can be obtained. The nanotube arrays showed in this manuscript are highly ordered anodized nanotubes obtained in HF electrolyte, which causes the serious corrosion on the nanotube array surface and nanotube walls. So far, anodization can only prepare TiO₂ nanotubes with rough uniform size, but can not achieve the nanotubes with identical size.

2. "Fluorine is good for life activity. The source of Fluorine is HF. HF is very toxic. How did authors make sure of complete removal of HF from TiO₂ nanotubes?"

--- We thank the Reviewer for the suggestion. After preparation process, the sample surface was rinsed repeatedly by distilled water, which is described by the sentence "The anodized samples are washed with distilled water" (line 13-14, 1st paragraph, section 2.1). Also, we have added the word "repeatedly" into this sentence "The anodized samples are washed with distilled water repeatedly" (line 14, 1st paragraph, section 2.1). Before biological experiments, all nanotube samples were ultrasonically cleaned for half an hour, and then immersed into distilled water for 24 hours to minimize HF residues in the interior of nanotubes and crevices among nanotubes, so as to eliminate the adverse effects of HF on cells. Also, we have added the sentence "In this work, all TiO₂ nanotube samples for biological experiments were ultrasonically cleaned for half an hour in distilled water and then were immersed into the distilled water for 24 hours to minimize HF electrolyte residues in the interior of nanotubes and crevices among nanotubes." in the biological experiments (Line 4-7, 3rd paragraph, section 2.3.1).

3. "The adhesion property of TiO₂ is not clearly explained in the manuscript."

--- We thank the Reviewer for the suggestion. We did not measure the adhesion strength of TiO₂ nanotubes on titanium substrate according to the standard method in this work. But after half an hour of ultrasonic cleaning, the TiO₂ nanotube arrays were not peeled off from the substrate, which indicated that anodized TiO₂ nanotube arrays possess excellent adhesion strength on the titanium substrate.

4. "It would be more interesting if authors compare the wettability and adhesion property of amorphous TiO₂ with crystalline TiO₂ nanotube."

--- We thank the Reviewer for the suggestion. We will compare the surface physicochemical properties, including the wettability and adhesion property, of amorphous TiO₂ nanotube arrays with crystalline TiO₂ nanotube arrays in our subsequent articles.

Thanks very much for your attention, and we hope to hear a positive response.

Jiahua Ni, Ph.D.

State Key Laboratory of Metal Matrix Composites.

Research Assistant of Material Science

School of Materials Science and Engineering.

Shanghai Jiao Tong University, Shanghai 200240, PR China.

Tel: 86 21-3420-2759, Fax: 86 21-3420-2759, E-mail: jiahua.ni@sjtu.edu.cn

**